# Evaluation of the Risk of *Clostridium difficile* Infection Using a Serum Bile Acid Profile

**DOI:** 10.3390/metabo12040331

**Published:** 2022-04-06

**Authors:** Tadakuni Monma, Junichi Iwamoto, Akira Honda, Hajime Ueda, Fumio Kakizaki, Shoichiro Yara, Teruo Miyazaki, Tadashi Ikegami

**Affiliations:** 1Division of Gastroenterology and Hepatology, Tokyo Medical University Ibaraki Medical Center, Ibaraki 300-0395, Japan; monkuni2@tokyo-med.ac.jp (T.M.); akihonda@tokyo-med.ac.jp (A.H.); h-ueda@tokyo-med.ac.jp (H.U.); tamasan903@gmail.com (F.K.); tmuimcyr@gmail.com (S.Y.); ikegamit@tokyo-med.ac.jp (T.I.); 2Joint Research Center, Tokyo Medical University Ibaraki Medical Center, Ibaraki 300-0395, Japan; teruom@tokyo-med.ac.jp

**Keywords:** antibiotics, bile acids, biomarker, *Clostridium difficile*, HPLC-MS/MS, proton pump inhibitors

## Abstract

Since intestinal secondary bile acids (BAs) prevent *Clostridium difficile* infection (CDI), the serum BA profile may be a convenient biomarker for CDI susceptibility in human subjects. To verify this hypothesis, we investigated blood samples from 71 patients of the Division of Gastroenterology and Hepatology at the time of admission (prior to antibiotic use and CDI onset). Twelve patients developed CDI during hospitalization, and the other 59 patients did not. The serum unconjugated deoxycholic acid (DCA)/[DCA + unconjugated cholic acid (CA)] ratio on admission was significantly lower in patients who developed CDI than in patients who did not develop CDI (*p* < 0.01) and in 46 healthy controls (*p* < 0.0001). Another unconjugated secondary BA ratio, 3β-hydroxy (3βOH)-BAs/(3βOH + 3αOH-BAs), was also significantly lower in patients who developed CDI than in healthy controls (*p* < 0.05) but was not significantly different between patients who developed and patients who did not develop CDI. A receiver operating characteristic (ROC) curve determined a cut-off point of DCA/(DCA + CA) < 0.349 that optimally discriminated on admission the high-risk patients who would develop CDI (sensitivity 91.7% and specificity 64.4%). In conclusion, a decreased serum DCA/(DCA + CA) ratio on admission strongly correlated with CDI onset during hospitalization in patients with gastrointestinal and hepatobiliary diseases. Serum BA composition could be a helpful biomarker for predicting susceptibility to CDI.

## 1. Introduction

*Clostridium difficile* infection (CDI) is one of the most common hospital and antibiotic-associated infections [1]. CDI causes various clinical symptoms, including diarrhea with colitis, abdominal pain, and fever [2]. The prevalence of CDI has been increasing worldwide, and refractory or life-threatening severe CDI is also reported in Western countries [3]. Therefore, in addition to the prevention and treatment of CDI, it would be helpful to develop a method for screening high-risk hospitalized patients.

Bile acids (BAs) are the end products of cholesterol metabolism and perform many chemical, physiological, and pathophysiological functions. BAs are synthesized in the liver, and cholic acid (CA) and chenodeoxycholic acid (CDCA) conjugated with glycine or taurine are secreted into bile as primary BAs (Figure 1). In the intestine, microbial bile salt hydrolase deconjugates amino acids to form free CA and CDCA, and then BA 7α-dehydroxylating bacteria convert CA and CDCA to the secondary BAs deoxycholic acid (DCA) and lithocholic acid (LCA), respectively. Many reports show the relationship between BAs and CDI. For example, the secondary BAs DCA and LCA inhibit *Clostridium difficile* growth in vitro [4,5] and in vivo [6,7,8,9], and secondary BAs in stool are reduced in patients with CDI [10,11].

*Clostridium scindens*, one of the BA 7α-dehydroxylating intestinal bacteria, converts the primary BAs to secondary BAs, and is positively correlated with the resistance to CDI [12,13]. Dehydroxylation at the 7α-position is encoded by multi-step bile acid-inducible (bai) genes in a single bai operon [14,15]. Fecal *bai*CD gene abundance represents the amount of BA 7α-dehydroxylating bacteria and was significantly higher in CDI-negative stools than in CDI samples [16]. Furthermore, these 7α-dehydroxylating gut bacteria synthesize not only secondary BAs but also tryptophan-derived antibiotics and inhibit the growth of *Clostridium difficile* [17]. These results suggest that we may predict the risk of CDI by BA analysis of hospitalized patients.

Fecal BA composition is a potential biomarker for the prediction of CDI. Allegretti et al. [10] reported that the ratio of unconjugated fecal DCA to glycoursodeoxycholic acid (GUDCA) was a predictor of CDI recurrence. However, we recently showed that the unconjugated fecal DCA/(DCA + CA) ratio was the best predictor of fecal proportion of *Clostridium* subcluster XIVa that includes *Clostridium scindens* [18]. In addition, the unconjugated serum DCA/(DCA + CA) ratio is also a possible marker for the fecal C. subcluster XIVa fraction.

In hospitals, serum is easier to obtain from patients than stool. Therefore, we tried to predict the risk of CDI in patients with gastrointestinal or hepatobiliary diseases by analyzing serum BA composition. If the results confirm our hypothesis, the DCA/(DCA + CA) ratio could be an aid in predicting susceptibility to CDI. 

## 2. Results

### 2.1. Baseline Characteristics of the Patients Enrolled in This Study

We enrolled 71 patients who had been admitted to the Gastroenterology and Hepatology division due to high inflammatory responses in blood tests. Twelve patients developed CDI during hospitalization and the other 59 patients did not. The baseline characteristics of the patients are shown in Table 1. Of the 71 patients, 34 had hepato-biliary-pancreatic diseases, 21 had gastrointestinal diseases other than inflammatory bowel diseases (IBD), 5 had IBD, and the other 11 had pneumonia or pyelonephritis. None had taken antibiotics on admission. After admission to the hospital, 66 out of 71 patients were administered intravenous antibiotics. In addition, about half of the enrolled patients had a history of regular use of proton pump inhibitors (PPIs).

### 2.2. Serum BA Composition in CDI Patients

A total of 10 conjugated and 20 unconjugated (free) BAs were quantified and compared among the different groups. As shown in Table 2, CA, 3-dehydro-CA, and glycochenodeoxycholic acid (GCDCA) proportions in patients with CDI were significantly higher than in those without CDI. Compared to healthy controls, the proportion of total unconjugated BAs decreased significantly and total glycine conjugated BAs increased significantly in patients with CDI. However, we observed the same tendencies in patients without CDI.

### 2.3. Serum BA Transformation Markers in CDI Patients

To estimate the effects of BA transformation, we calculated the product/(product+substrate) ratio for the specific reactions that may be related to the inhibition of CD growth (Figure 1). We calculated BA deconjugation by free/total primary BAs, 7α-dehydroxylation of BAs by DCA/(DCA + CA) or LCA/(LCA + CDCA), and the epimerization of 3α-hydroxy-BAs (3αOH-BAs) to 3β-hydroxy-BAs (3βOH-BAs) by 3βOH-BAs/(3βOH + 3αOH-BAs). As shown in Figure 2, DCA/(DCA+CA) on admission (prior to antibiotic use and CDI onset) was significantly lower in patients who developed CDI during hospitalization than in patients who did not develop CDI ( *p* < 0.01) and healthy controls (*p* < 0.0001). Serum DCA levels were not necessarily decreased in patients with CDI (Table 2) because they are affected by the total amount of BAs in the colon and the rate of BA absorption from the colon. However, the product/(product + substrate) ratio represents the conversion rate from CA to DCA, which is not easily affected by conditions other than enzyme activity. Although 3βOH-BAs/(3βOH + 3αOH-BAs) in patients with CDI was not significantly different from that in patients without CDI, it was significantly lower than that in healthy controls (*p* < 0.05). Free/total primary BAs and LCA/(LCA + CDCA) were not significantly different among the groups.

### 2.4. Comparison of Serum BA Markers among the Underlying Diseases

On admission, the BA ratios were compared among the three patient groups, hepato-biliary-pancreatic diseases, gastrointestinal diseases (except for IBD), and IBD (Figure 3). The free/total primary BAs ratio was significantly lower in hepato-biliary-pancreatic diseases than in gastrointestinal diseases (*p* < 0.05). The DCA/(DCA + CA) ratio was significantly lower in IBD than in gastrointestinal diseases (*p* < 0.01). The LCA/(LCA + CDCA) and 3βOH-BAs/(3βOH + 3αOH-BAs) ratios were not significantly different among the groups.

### 2.5. Effects of Treatment with Antibiotics on Serum BA Markers

After hospitalization, 61 out of 71 patients were administered intravenous antibiotics. Six patients developed CDI during or after treatment, and the other 55 did not. Serum BA markers in the pair sera before and after antibiotics were analyzed. As shown in Figure 4, DCA/(DCA + CA) and 3βOH-BAs/(3βOH + 3αOH-BAs) ratios were significantly decreased by treatment with antibiotics (*p* < 0.0001). However, free/total primary BAs and LCA/(LCA + CDCA) ratios did not change significantly after using antibiotics.

### 2.6. Effects of the Use of PPIs on Serum BA Markers

There were no significant differences in free/total primary BAs or the DCA/(DCA + CA), LCA/(LCA + CDCA), or 3βOH-BAs/(3βOH + 3αOH-BAs) ratios between patients who did and did not take PPIs (including a new potassium-competitive acid blocker, vonoprazan) (Figure 5).

### 2.7. The Receiver Operating Characteristic (ROC) Analyses for the Prediction of CDI Development by Serum BA Markers

We calculated the sensitivity and specificity of each BA marker to predict CDI development using 71 patients by ROC analyses (Figure 6). The areas under the curve (AUC) and the 95% confidence intervals of free/total primary BAs, DCA/(DCA + CA), LCA/(LCA + CDCA), and 3βOH-BAs/(3βOH + 3αOH-BAs) ratios were 0.6045 (0.4648–0.7443) (NS), 0.7571 (0.6217–0.8924) (*p* < 0.01), 0.5240 (0.3333–0.7147) (NS), and 0.6681 (0.5248–0.8113) (*p* = 0.068), respectively. Since DCA/(DCA + CA) had the largest AUC, this ratio appears to be the optimal biomarker for the prediction of CDI. The cut-off value of DCA/(DCA + CA) was <0.349 for discriminating the high-risk patients with CDI on admission (prior to antibiotic use and CDI onset). At this value, the sensitivity was 91.67%, the specificity was 66.10%, and the likelihood ratio was 2.704.

## 3. Discussion

Our results demonstrated that the blood proportion of secondary BAs was a suitable biomarker to identify patients at high risk of developing CDI during hospital admission. Many reports show the relationship between BAs and CDI, but only a single limited study [10] has utilized BA composition as a surrogate marker for predicting susceptibility to CDI. The unconjugated fecal DCA/GUDCA ratio was reported to be a predictor of the recurrence of CDI. However, GUDCA is not a substrate for DCA, and the concentration of GUDCA is affected by many factors, including conversion from CDCA to UDCA in the colon, absorption from the colon, glycine conjugation in the liver, deconjugation in the intestine, and the possibility of administration in patients with hepatobiliary diseases. In contrast, as a predictor of CDI, we used DCA/(DCA + CA), which is a product/(product+substrate) ratio, to calculate dehydroxylating activity at the 7α-position of CA. Furthermore, we determined the ratio not in the stool, but in the serum. Our previous data showed that the fecal proportion of *Clostridium* subcluster XIVa correlated better with the DCA/(DCA + CA) in the feces than in the serum [18]. However, a patient’s serum is more easily obtainable than stool in hospital. When predicting intestinal 7α-dehydroxylating activity using serum BA profile, it is essential to use only deconjugated DCA and CA for calculation. Only deconjugated CA is transformed to DCA by intestinal bacteria, and almost all deconjugated CA and DCA absorbed from the intestine are re-conjugated with glycine or taurine in the liver. Therefore, among serum BAs, only the deconjugated BAs directly reflect the activity of secondary BA production in the intestine.

In addition to DCA/(DCA + CA), we also calculated LCA/(LCA + CDCA), 3βOH-BAs/(3βOH + 3αOH-BAs), and free/total primary BAs in this study. LCA/(LCA + CDCA) should also reflect the 7α-dehydroxylating activity of the intestinal bacteria but showed different results from DCA/(DCA + CA) (Figure 2, Figure 3 and Figure 4). In healthy subjects, the LCA/(LCA + CDCA) ratios were much smaller than the DCA/(DCA + CA) ratios in serum but not in feces [18], which suggests that LCA is less readily absorbed from the intestine than other BAs. Therefore, DCA/(DCA + CA) is a better serum marker for 7α-dehydroxylation than LCA/(LCA + CDCA). Serum 3βOH-BAs/(3βOH + 3αOH-BAs) represents epimerization activity from 3αOH to 3βOH. In our previous study [18], this ratio was also positively correlated with the fecal fraction of *Clostridium* subcluster XIVa. Although it is not clear if this bacterial subcluster epimerizes the hydroxyl group at the C-3 position, the change in 3βOH-BAs/(3βOH + 3αOH-BAs) was associated with the change in DCA/(DCA + CA) (Figure 2, Figure 3 and Figure 4). On the other hand, free/total primary BAs may be a surrogate marker for small intestinal bacterial overgrowth (SIBO). In bile, almost all BAs are conjugated with amino acid and deconjugated by bile salt hydrolases of various genera in the gut microbiota, including *Bacteroides, Bifidobacterium, Clostridium, Enterococcus*, and *Lactobacillus* [19]. Since nearly 95% of BAs are reabsorbed from the small intestine [14], and most primary BAs originate from the small rather than the large intestine, patients with SIBO may have an increased serum deconjugated (free) primary BA fraction.

At the time of admission (prior to antibiotic use and CDI onset), we measured the above BA markers in patients who were admitted to the Gastroenterology and Hepatology division due to high inflammatory responses in blood tests. Our results demonstrated that DCA/(DCA + CA) on admission was significantly lower in patients who developed CDI during hospitalization than in patients who did not develop CDI and healthy controls (Figure 2). In addition, ROC analyses showed that DCA/(DCA + CA) had the largest AUC, indicating that this ratio is the optimal biomarker for the prediction of CDI development (Figure 5). The cut-off value of DCA/(DCA + CA) on admission for discriminating patients at high risk of developing CDI was <0.349. At this value, the sensitivity and specificity were 91.67% and 66.10%, respectively. Therefore, patients with a DCA/(DCA + CA) of less than 0.349 at the time of admission should be monitored for CDI development during hospitalization.

In many cases of high inflammatory responses in blood tests, antibiotics are used after hospitalization. In fact, 66 out of 71 patients were treated with antibiotics after hospitalization (Table 1). The antibiotics caused dysbiosis with decreased DCA/(DCA + CA) and 3βBA/(3βBA + 3αBA) (Figure 4). However, our results suggest that most patients who developed CDI were already in dysbiosis on admission (Figure 2). In particular, IBD patients had the lowest DCA/(DCA + CA) ratio on admission (Figure 3). We have already reported that DCA/(DCA+CA) in feces and serum is decreased in IBD patients, regardless of disease activity [18]. Although the mechanism of dysbiosis in IBD patients is not fully understood, IBD is considered a significant risk factor for CDI development. While most patients developed CDI after the use of antibiotics, some IBD patients developed CDI without antibiotics. These IBD patients were hospitalized due to worsening of their primary disease, but we cannot exclude the possibility that they had already developed CDI at the time of admission. However, in any case, patients with low DCA/(DCA + CA) on admission remain a risk group for CDI. A recent study by Berkell et al. [20] showed that patients developing CDI already exhibited distinct microbiota and significantly lower diversity before antibiotic treatment, suggesting the possibility of a predictive microbiota-based diagnosis of CDI. Our results indicate that not only microbiota-based diagnostics but also serum BA composition, such as DCA/(DCA + CA), could be convenient predictive markers for CDI.

In addition to the impacts of antibiotics and IBD, we examined the effects of PPIs on the BA markers. Previous reports showed that the use of PPIs altered the composition of gut microbiota significantly, more than the use of antibiotics or other drugs [21,22]. As a result, PPI users have an increased risk of CDI. However, our data showed that any BA markers were not significantly different between patients with and without PPI treatment (Figure 5). Therefore, the use of PPIs does not appear to be the primary cause of CDI development in our patients.

There are several limitations to this study. First, as the sample size of CDI patients was small, a multicenter study with a large sample size is needed to validate the results. Second, since most of the patients enrolled in this study had gastrointestinal or hepatobiliary diseases, further studies are needed using patients other than those with digestive disorders. Third, multivariate analysis with other “classic” CDI risk factors would be essential to validate the utility of our new biomarker.

In conclusion, decreased serum DCA/(DCA + CA) on admission in patients admitted to the Gastroenterology and Hepatology division due to high inflammatory responses in blood tests exhibits a strong correlation with a high risk of CDI development during hospitalization. Thus, serum BA profile, especially decreased serum DCA/(DCA + CA), could be a convenient surrogate marker for the prediction of CDI development.

## 4. Materials and Methods

### 4.1. Sample Collection

Fasting blood samples were collected from patients at the time of admission (prior to antibiotic use and CDI onset). Fasting blood samples were also obtained from 46 healthy volunteers (35 males and 11 females, aged 47.6 ± 8.2 years). Sera were stored at −20 °C until analysis.

### 4.2. Diagnosis of CDI

During hospitalization, 12 patients developed frequent diarrhea and were diagnosed with CDI according to a flow chart by Czepiel et al. [2]. Fecal *Clostridium difficile*-specific glutamate dehydrogenase (GDH) and toxins (CD toxins) were determined by GE test immunochromato-CD GDH/TOX “NISSUI” (Nissui Pharmaceutical Co., LTD., Tokyo, Japan) and used for the CDI screening test.

### 4.3. Serum BA Analyses

Serum BA compositions were measured by HPLC-MS/MS as described by Murakami et al. [18]. Briefly, a mixture of internal standards was added to 20 µL of serum and diluted with 2 mL of 0.5 M potassium phosphate buffer (pH 7.4). BAs were extracted with Bond Elut C18 cartridges and analyzed by HPLC-MS/MS system.

### 4.4. Statistical Analysis

Data are reported as the mean ± SEM. The statistical significance of differences among the three groups was evaluated by the Tukey–Kramer test, the difference between the two groups was by the Mann-Whitney test, and the difference before and after treatment was by paired t-test. Categorical variables were analyzed using Fisher’s exact test. The ROC curve was used for the analysis of the values of free/total primary BAs, DCA/(DCA + CA), LCA/(LCA + CDCA), and 3βOH-BAs/(3βOH + 3αOH-BAs) in the prediction of CDI. The minimum distance from the upper left corner (0, 1) was considered the optimal cut-off value. Sensitivity was calculated as true positive number/(true positive number + false negative number), and specificity was as true negative number/(true negative number + false positive number). For all analyses, significance was accepted at the level of *p* < 0.05. All statistical analyses were conducted using Prism (ver. 9.2.0) software (GraphPad Software, San Diego, CA, USA).

## Figures and Tables

**Figure 1 metabolites-12-00331-f001:**
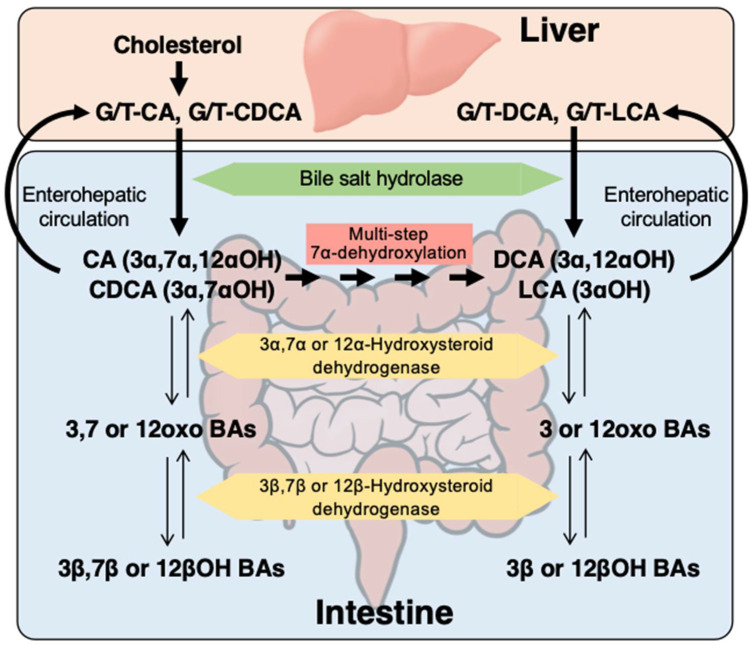
Metabolism of primary bile acids (BAs) conjugated with amino acid (glycine or taurine). Microbial bile salt hydrolase deconjugates the amino acid to form free cholic acid (CA) and chenodeoxycholic acid (CDCA). The free (deconjugated) CA and CDCA are then metabolized to deoxycholic acid (DCA) and lithocholic acid (LCA), respectively by multi-step 7α-dehydroxylation. Hydroxyl groups at the 3α, 7α, and 12α positions can be converted to carbonyl groups by 3α-, 7α-, and 12α-hydroxysteroid dehydrogenases, respectively. In addition, the carbonyl groups at the 3, 7, and 12 positions can be converted to hydroxyl groups at the 3β, 7β, and 12β positions by the reverse reactions of 3β-, 7β-, and 12β-hydroxysteroid dehydrogenases, respectively. G/T-, glycine or taurine conjugated.

**Figure 2 metabolites-12-00331-f002:**
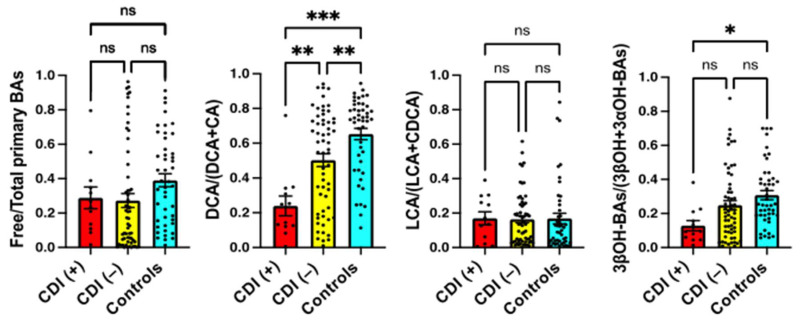
Serum BA markers in patients who were admitted to the Gastroenterology and Hepatology division due to high inflammatory responses in blood tests. Serum samples were obtained on admission (prior to antibiotic use and CDI onset). CDI (+), patients who developed CDI during hospitalization (*n* = 12); CDI (−), patients who did not develop CDI (*n* = 59); Controls, healthy controls (*n* = 46). Each column and error bar represents the mean and SEM. According to the Tukey–Kramer test, * *p* < 0.05, ** *p* < 0.01, and *** *p* < 0.0001 were significantly different. ns, not significant.

**Figure 3 metabolites-12-00331-f003:**
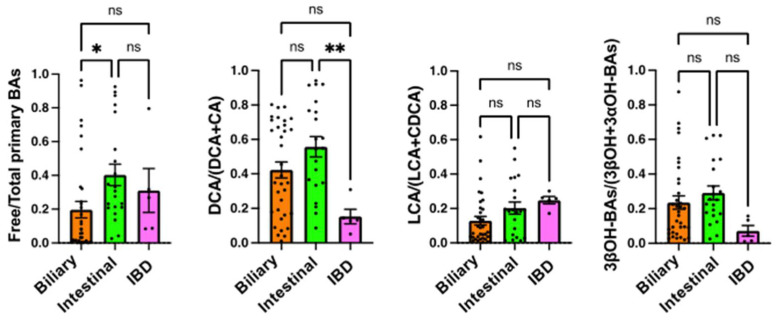
Comparison of serum BA markers among patients with different underlying diseases. Serum samples were obtained on admission (prior to antibiotic use and CDI onset). Biliary, patients with hepato-biliary-pancreatic diseases (*n* = 34); Intestinal, patients with gastrointestinal diseases (*n* = 21); IBD, patients with inflammatory bowel diseases (*n* = 5). Each column and error bar represents the mean and SEM. According to the Tukey–Kramer test, * *p* < 0.05 and ** *p* < 0.01 were significantly different; ns, not significant.

**Figure 4 metabolites-12-00331-f004:**
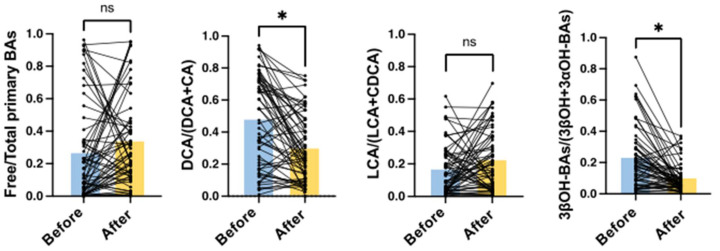
Effects of treatment with antibiotics on serum BA markers. Serum samples were obtained on admission (prior to antibiotic use and CDI onset) and after intravenous administration of antibiotics (*n* = 61). Before, before using antibiotics; After, after using antibiotics. The mean value for each group is indicated by the columns. According to a paired *t*-test, * *p* < 0.0001 was significantly different; ns, not significant.

**Figure 5 metabolites-12-00331-f005:**
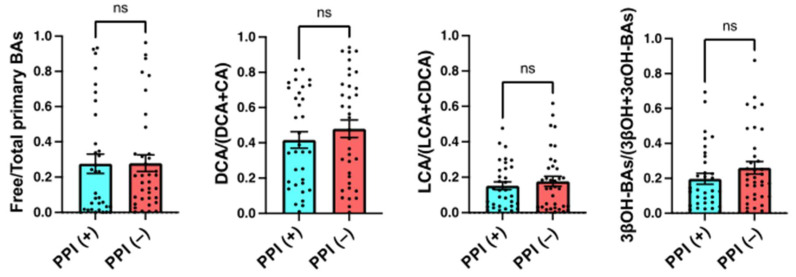
Effects of proton pump inhibitor use on serum BA markers. Serum samples were obtained on admission (prior to antibiotic use and CDI onset). PPI (+), patients using proton pump inhibitors (*n* = 33); PPI (−), patients not using proton pump inhibitors (*n* = 36). Each column and error bar represents the mean and SEM. Statistical significance was tested by the Mann–Whitney test; ns, not significant.

**Figure 6 metabolites-12-00331-f006:**
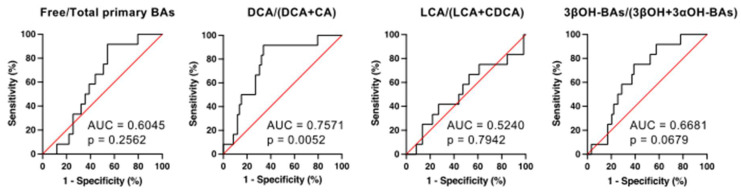
ROC analyses for the prediction of CDI development by serum BA markers. AUC, area under the curve. Sensitivity = true positive number/(true positive number + false negative number); specificity = true negative number/(true negative number + false positive number). The minimum distance from the upper left corner (0, 1) was considered the optimal cut-off value. The cut-off value of DCA/(DCA+CA) was <0.349 in discriminating the high-risk patients with CDI before treatment with antibiotics. At this value, the sensitivity was 91.67%, the specificity was 66.10%, and the likelihood ratio was 2.704.

**Table 1 metabolites-12-00331-t001:** Characteristics of enrolled patients with and without *Clostridium difficile* infection (CDI).

	CDI (+) (*n* = 12)	CDI (−) (*n* = 59)	*p*-Value
Age (mean ± SEM)	63.2 ± 26.5	69.0 ± 19.0	*NS*
Gender (male:female)	10:2	34:25	*NS*
Underlying diseases			
Hepato-biliary-pancreatic diseases *^a^*	3	31	*NS*
Gastrointestinal diseases *^b^*	2	19	*NS*
Inflammatory bowel disease *^c^*	4	1	*p* < 0.01
Others *^d^*	3	8	*NS*
Medications			
Antibiotics (on admission)	0	0	*NS*
Antibiotics (after admission) *^e^*	7	59	*p* < 0.001
PPIs including vonoprazan (regular use)	7	26 ^*f*^	*NS*

CDI (+), patients who developed CDI during hospitalization; CDI (−), patients who did not develop CDI; SEM, standard error of the mean; PPIs, proton pump inhibitors. *^a^* Hepato-biliary-pancreatic diseases include cholangitis, cholecystitis, liver abscess, and pancreatitis. *^b^* Gastrointestinal diseases include infectious enteritis, diverticulitis of colon, ileus, and acute appendicitis, and do not include inflammatory bowel diseases. *^c^* Inflammatory bowel diseases include ulcerative colitis and Crohn’s disease. *^d^* Others include pneumonia and pyelonephritis. *^e^* Antibiotics include ceftizoxime sodium, cefmetazole sodium, piperacillin/tazobactam, levofloxacin hydrate, and meropenem hydrate. ^*f*^ Two of 59 CDI (−) patients had an unknown history of taking PPIs. *p* < 0.05 was significantly different according to Fisher’s exact test. *NS*, not significant.

**Table 2 metabolites-12-00331-t002:** Serum BA composition in subjects with and without *Clostridium difficile* infection (CDI).

	CDI (+) (*n* = 12)	CDI (−) (*n* = 59)	Controls (*n* = 46)
	%	%	%
**Unconjugated BAs**	**32.51 ± 7.05 ^†^**	**35.64 ± 4.10** ** ^†^ **	**55.97 ± 3.65**
CA	11.93 ± 2.95 ** ^†^	4.36 ± 0.65	6.02 ± 0.85
CDCA	5.33 ± 1.08	6.64 ± 1.27	10.71 ± 1.27
DCA	5.51 ± 2.38	7.77 ± 1.30 ^†^	12.88 ± 1.52
LCA	1.30 ± 0.47	0.63 ± 0.09 ^†^	1.33 ± 0.24
UDCA	1.26 ± 0.71	3.57 ± 0.89	2.99 ± 0.36
UCA	0.23 ± 0.09	0.11 ± 0.02	0.19 ± 0.03
3-epi-CA	0.30 ± 0.09	0.26 ± 0.06	0.20 ± 0.04
12-epi-CA	0.11 ± 0.05	0.03 ± 0.01 ^†^	0.11 ± 0.02
3-epi-DCA&CDCA	1.58 ± 0.80 ^†^	1.76 ± 0.29 ^††^	4.12 ± 0.39
3-epi-UDCA	1.814 ± 0.85 ^†^	7.27 ± 1.33	12.10 ± 1.93
3-epi-LCA	0.35 ± 0.11	0.15 ± 0.03 ^††^	0.60 ± 0.09
3-dehydro-CA	0.32 ± 0.13 * ^†^	0.10 ± 0.02	0.15 ± 0.03
7-oxo-DCA	0.35 ± 0.13	0.34 ± 0.11	0.23 ± 0.08
12-oxo-CDCA	0.23 ± 0.11	0.14 ± 0.03	0.14 ± 0.03
3-dehydro-CDCA	0.34 ± 0.19 ^†^	0.22 ± 0.05 ^††^	0.97 ± 0.14
3-dehydro-DCA	0.63 ± 0.22	0.72 ± 0.17	0.65 ± 0.11
3-dehydro-UDCA	0.22 ± 0.09	0.41 ± 0.10	0.37 ± 0.06
7-oxo-LCA	0.22 ± 0.09	0.25 ± 0.05 ^†^	0.63 ± 0.10
12-oxo-LCA	0.29 ± 0.12	0.78 ± 0.25	0.98 ± 0.18
3-dehydro-LCA	0.18 ± 0.09 ^†^	0.15 ± 0.04 ^††^	0.61 ± 0.08
**Glycine conjugated BAs**	**59.83 ± 8.08 ^†^**	**51.53 ± 3.19 ^†^**	**40.45 ± 3.07**
GCA	14.84 ± 4.59	13.91 ± 2.23 ^†^	7.02 ± 1.44
GCDCA	32.17 ± 5.97 * ^†^	21.69 ± 1.64	20.03 ± 1.59
GDCA	3.81 ± 1.46	6.90 ± 1.06	8.21 ± 1.08
GLCA	0.02 ± 0.01	0.21 ± 0.07	0.10 ± 0.03
GUDCA	8.99 ± 6.53	8.83 ± 1.51	5.10 ± 0.74
**Taurine conjugated BAs**	**7.66 ± 2.79**	**12.83 ± 2.03 ^††^**	**3.58 ± 0.88**
TCA	0.99 ± 0.32	2.58 ± 0.61	1.09 ± 0.36
TCDCA	4.06 ± 1.28	8.43 ± 1.44 ^††^	1.90 ± 0.47
TDCA	2.30 ± 1.85	1.56 ± 0.32	0.53 ± 0.15
TLCA	0.05 ± 0.02	0.02 ± 0.01	0.03 ± 0.01
TUDCA	0.26 ± 0.14	0.23 ± 0.08	0.03 ± 0.01
**Total BAs**	**100**	**100**	**100**

CDI (+), patients who developed CDI during hospitalization; CDI (−), patients who did not develop CDI; Controls, healthy volunteers; CA, cholic acid; CDCA, chenodeoxycholic acid; DCA, deoxycholic acid; LCA, lithocholic acid; UDCA, ursodeoxycholic acid; G, glyco (glycine conjugated); T, tauro (taurine conjugated). Each entry in the table represents the mean ± SEM. According to the Tukey–Kramer test, * *p* < 0.05 and ** *p* < 0.001 were significantly different from CDI (−). According to the Tukey–Kramer test, ^†^
*p* < 0.05 and ^††^
*p* < 0.001 were significantly different from controls.

## Data Availability

Not applicable.

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
