# Peer review of "Evaluation of the Risk of Clostridium difficile Infection Using a Serum Bile Acid Profile"

_metabolites, 2022, doi:10.3390/metabo12040331_

Round 1
Reviewer 1 Report
Introduction
Line 62. Please, consider to change the sentence: “The results showed that we could predict CDI susceptibility to some extent by determining serum unconjugated DCA/(DCA+CA) ratio” to this one: “If the results confirm our hypothesis, the DCA/(DCA+CA) ratio could be an aid in predicting susceptibility to CDI”.
Results
There are acronyms such as TUDCA or TLCA that are not defined or explained anywhere.
Tables
Table 2. Regarding the p value of the tables, I think it would be better to use two categories: T < 0.05 and TT < 0.001. I do not think it is of great interest to make 4 categories as there currently are.
Figures
In figure 4, it is recommended to change ****p < 0.0001 by * p<0.001 (I think that since it is an independent figure, it is not necessary to use the 4 asterisks but 1).
Discussion
I think the first paragraph of the discussion should be devoted to commenting on the fundamental findings in the study (in this case that the blood concentration of secondary bile acids is a suitable biomarker to identify patients at high risk of developing DCI during hospital admission).
As a suggestion, I think the authors should comment if they consider interesting to perform a future study in which the relationship between the blood concentration of secondary bile acids and the risk of CDI would be adjusted, but introducing in a multivariate analysis "classic" risk factors such as age, previous episodes of CRF, severe episodes of CDI, renal failure or lack of initial response to treatment, among others).
The "conclusions" paragraph should be preceded by a paragraph on the limitations of the study.
Author Response
Replies to the comments from reviewer #1
- Introduction. Line 62. Please, consider to change the sentence: “The results showed that we could predict CDI susceptibility to some extent by determining serum unconjugated DCA/(DCA+CA) ratio” to this one: “If the results confirm our hypothesis, the DCA/(DCA+CA) ratio could be an aid in predicting susceptibility to CDI”.
RESPONSE: Thank you for the reviewer’s suggestion. We have changed the Line 62 as indicated by the reviewer.
- Results. There are acronyms such as TUDCA or TLCA that are not defined or explained anywhere.
RESPONSE: We are using TUDCA and TLCA in Table 2. We have added explanations for CA, CDCA, DCA, LCA, UDCA, T (taurine conjugated), and G (glycine conjugated) in the footnote of Table 2.
- Tables. Table 2. Regarding the p value of the tables, I think it would be better to use two categories: T < 0.05 and TT < 0.001. I do not think it is of great interest to make 4 categories as there currently are.
RESPONSE: Thank you for pointing this out. We agree with the reviewer and have changed the categories regarding the p value of the Table 2 from three or four to two.
- Figures. In figure 4, it is recommended to change ****p < 0.0001 by * p<0.001 (I think that since it is an independent figure, it is not necessary to use the 4 asterisks but 1).
RESPONSE: We have changed the asterisks of p value in figures 2 and 4 as suggested by the reviewer.
- Discussion. I think the first paragraph of the discussion should be devoted to commenting on the fundamental findings in the study (in this case that the blood concentration of secondary bile acids is a suitable biomarker to identify patients at high risk of developing DCI during hospital admission).
RESPONSE: We appreciate the reviewer’s suggestion. We have added the following sentence at the beginning of the Discussion. Our results demonstrated that the blood proportion of secondary bile acids was a suitable biomarker to identify patients at high risk of developing CDI during hospital admission.
- As a suggestion, I think the authors should comment if they consider interesting to perform a future study in which the relationship between the blood concentration of secondary bile acids and the risk of CDI would be adjusted, but introducing in a multivariate analysis "classic" risk factors such as age, previous episodes of CRF, severe episodes of CDI, renal failure or lack of initial response to treatment, among others).
The "conclusions" paragraph should be preceded by a paragraph on the limitations of the study.
RESPONSE: We thank the reviewer’s comment that multivariate analysis, including "classic" risk factors, is interesting as a future study. We have added a paragraph on the limitations of this study in which we mentioned sample size, background diseases, and the relation to "classic" risk factors.

Reviewer 2 Report
Reviewer’s comment
In the study titled ‘Evaluation of the risk of Clostridium difficile infection using a 2 serum bile acid profile’, the authors Monma et al., have used serum-bile acid profile to predict Clostridium difficile infection (CDI). BA is reported to inhibit the germination of C. difficile spores. The authors have done the bile acid (BA) profiling of serum collected from 71 patients admitted to Gastroenterology and Hepatology of Tokyo Medical University Ibaraki Medical Center, Ibaraki, Japan. Blood samples were collected before antibiotic treatment. Additionally, blood samples were collected from healthy volunteers. Twelve patients developed CDI during hospitalization, and the other 59 patients did not. Serum-BA and metabolites were measured by HPLC-MS/MS. Among many BA and its metabolites analyzed, the ratio of unconjugated deoxycholic acid (DCA)/[DCA+unconjugated cholic acid (CA)] ratio on admission was significantly lower in patients who developed CDI than in patients who did not develop CDI. A cut-off point of DCA/(DCA+CA) < 0.349 was determined using a receiver operating characteristic (ROC) curve. This cut-off value optimally discriminated on the admission of the high-risk 23 patients who would develop CDI (sensitivity 91.7% and specificity 64.4%).
This study is a follow-up to the previous study by Murakami et al 2018 (Inflamm Bowel Dis ). The manuscript is well written. Here are a few suggestions that might improve the value of the manuscript.
- The introduction can be more descriptive, especially the metabolism of bile acid by the microbiome.
- In Line 71-72, It is mentioned that none had taken antibiotics on admission; hence in table-1, it is good to mention that the antibiotic was administered after admission to the hospital as stated in line 145.
- The authors may consider including the study by Berkell et al. 2021 (https://www.nature.com/articles/s41467-021-22302-0) in the discussion.
- Discussion: The authors may consider including the sample size as one of the limitations of the study.
- Methodology for CDI detection to be included.

Author Response
Replies to the comments from reviewer #2
- The introduction can be more descriptive, especially the metabolism of bile acid by the microbiome.
RESPONSE: Thank you for the reviewer’s comment. Metabolism of bile acids by the microbiome has been added to the second paragraph of the Introduction. In addition, we have moved Fig. 1 to the Introduction.
- In Line 71-72, It is mentioned that none had taken antibiotics on admission; hence in table-1, it is good to mention that the antibiotic was administered after admission to the hospital as stated in line 145.
RESPONSE: We have revised Table 1 and added the sentence that the antibiotic was administered after admission to the hospital to the end of section 2.1. Also, we have added the explanation of PPIs.
- The authors may consider including the study by Berkell et al. 2021 (https://www.nature.com/articles/s41467-021-22302-0) in the discussion.
RESPONSE: Thank you for the reviewer’s suggestion. We have cited Berkell’s paper [20] at the end of the 4th paragraph of the Discussion.
- Discussion: The authors may consider including the small sample size as one of the limitations of the study.
RESPONSE: The reviewer is right. We have added the paragraph about the limitations, including the small sample size, to the end of the Discussion.
- Methodology for CDI detection to be included.
RESPONSE: We have added the CDI detection method in section 4.2 of the Materials and Methods.

Reviewer 3 Report
Monma et al. suggest serum DCA/(DCA+CA) ratio as a marker to predict CDI susceptibility. While I am convinced that the serum DCA/(DCA+CA) ratio can be used as a clinical marker to predict susceptibility, I am still concerned about the way of interpretation of the dataset.
The authors declare that the serum DCA/(DCA+CA) ratio is the way of calculating the 7a-hydroxylating activity of intestinal bacteria. However, in Table 2, the DCA level between CDI+ and CDI- is not significantly different; rather CA level seems to dominate the result of the DCA/(DCA+CA) ratio. Even though CA is the substrate for DCA, if the authors want to declare reduction of DCA/(DCA+CA) ratio in CDI+ patients is due to the 7a-hydroxylating activity, the author should have been able to see a significant decrease in DCA levels. I recommend that the authors thoroughly consider this fact and describe it more clearly throughout the text and discussion.
Author Response
Replies to the comments from reviewer #3
Monma et al. suggest serum DCA/(DCA+CA) ratio as a marker to predict CDI susceptibility. While I am convinced that the serum DCA/(DCA+CA) ratio can be used as a clinical marker to predict susceptibility, I am still concerned about the way of interpretation of the dataset.
The authors declare that the serum DCA/(DCA+CA) ratio is the way of calculating the 7a-hydroxylating activity of intestinal bacteria. However, in Table 2, the DCA level between CDI+ and CDI- is not significantly different; rather CA level seems to dominate the result of the DCA/(DCA+CA) ratio. Even though CA is the substrate for DCA, if the authors want to declare reduction of DCA/(DCA+CA) ratio in CDI+ patients is due to the 7a-hydroxylating activity, the author should have been able to see a significant decrease in DCA levels. I recommend that the authors thoroughly consider this fact and describe it more clearly throughout the text and discussion.
RESPONSE: Thank you for the reviewer’s critical comment. As indicated by the reviewer, the DCA level in the colon should be decreased when bacterial 7alpha-dehydroxylating activity is reduced. However, serum DCA levels are not necessarily decreased in these patients because they are affected by the total amount of BAs in the colon and the rate of BA absorption from the colon. To offset these effects, we calculated the product/(product+substrate) ratio, which represents the conversion rate from CA to DCA, unaffected by conditions other than enzyme activity. We have explained the difference between DCA level and DCA/(DCA+CA) ratio in section 2.3 of the Results.

Round 2
Reviewer 3 Report
The authors well addressed my concern.